# Classification and Regression Trees analysis identifies patients at high risk for kidney function decline following hospitalization

**Weihao Wang**[1], **Wei Zhu**[1], **Janos Hajagos**[2], **Laura Fochtmann**[2,3], **Farrukh M. Koraishy**[4]*

**1** Department of Applied Mathematics, Stony Brook University, Stony Brook, NY, United States of America, **2** Department of Biomedical Informatics, Stony Brook University, Stony Brook, NY, United States of America, **3** Department of Psychiatry, Stony Brook University, Stony Brook, NY, United States of America, **4** Division of Nephrology, Department of Medicine, Stony Brook University, Stony Brook, NY, United States of America

* Farrukh.Koraishy@stonybrookmedicine.edu

**Data Availability Statement:** All relevant data are within the paper and its Supporting Information files.

## Abstract

Estimated glomerular filtration rate (eGFR) decline is associated with negative health outcomes, but the use of decision tree algorithms to predict eGFR decline is underreported. Among patients hospitalized during the first year of the COVID-19 pandemic, it remains unclear which individuals are at the greatest risk of eGFR decline after discharge. We conducted a retrospective cohort study on patients hospitalized at Stony Brook University Hospital in 2020 who were followed for 36 months post discharge. Random Forest (RF) identified the top ten features associated with fast eGFR decline. Logistic regression (LR) and Classification and Regression Trees (CART) were then employed to uncover the relative importance of these top features and identify the highest risk patients. In the cohort of 1,747 hospital survivors, 61.6% experienced fast eGFR decline, which was associated with younger age, higher baseline eGFR, and acute kidney injury (AKI). Multivariate LR analysis showed that older age was associated with lower odds of fast eGFR decline whereas length of hospitalization and vasopressor use with greater odds. CART analysis identified length of hospitalization as the most important factor and that patients with AKI and hospitalization of 27 days or more were at highest risk. After grouping by ICU and COVID-19 status and propensity score matching for demographics, these risk factors of fast eGFR decline remained consistent. CART analysis can help identify patient subgroups with the highest risk of post-discharge eGFR decline. Clinicians should consider the length of hospitalization in post-discharge monitoring of kidney function.

## Introduction

Fast decline in kidney function, as measured by estimated glomerular filtration rate (eGFR), is a hallmark of the development and progression of chronic kidney disease (CKD), and associated with adverse health outcomes [1–4]. Inadequate kidney recovers after acute kidney injury (AKI), a common diagnosis in hospitalized patients, is strongly associated with the risk of

**Funding:** The author(s) received no specific funding for this work.

**Competing interests:** The authors have declared that no competing interests exist.

CKD [5]. Among hospitalized patients, post-discharge eGFR decline is seen in patients with or without AKI and associated with increased risk of death [6–8].

Machine learning (ML) models have been used to predict the risk of kidney disease [9–11]. In studies of patients with CKD, logistic regression (LR) and random forest (RF) algorithms have been used to identify patients at risk of fast eGFR decline [12, 13]. Although predictive models of eGFR decline identify important clinical factors, the relative importance of these factors and the combination of factors that identify highest and lowest risk patient subgroups is less extensively studied. Classification and Regression Trees (CART) is a non-parametric and non-linear supervised ML algorithm that identifies the various factors associated with clinical outcomes in an open-box decision tree approach. The feature importance of each node (clinical variable) is determined by its hierarchy in the decision tree. The resulting cluster and hierarchy of nodes can identify the patient subgroups at lowest and highest risk of the clinical outcome. CART has been used in the study of kidney diseases including AKI [14–17], urinary obstruction [18], and dialysis patients [19]. Although decision tree algorithms have also been used to predict CKD [20–26], their use in the prediction of eGFR decline has rarely been reported [27, 28]. The only study that clearly identified the use of CART to predict eGFR decline, was limited to a small, select group of patients and did not investigate post-hospitalization eGFR decline [28].

We recently reported the characteristics of Coronavirus disease 2019 (COVID-19) associated AKI in hospitalized patients in a large national cohort of 336,473 patients with COVID-19, out of whom 129,176 (38%) patients had AKI [29]. In a previous study at our medical center, we reported that among hospitalized patients, the prevalence of AKI was 41.3% in those with COVID-19, while only 24.2% in those without COVID [30]. Although COVID-19 associated AKI was recently reported to be associated with a lower risk of kidney outcomes compared to AKI due to other causes [31], the study of post-discharge kidney function in those without AKI is lacking. There is also lack of data on the use of ML models to predict factors associated with eGFR decline in patients hospitalized with COVID-19.

We hypothesized that using CART, we will be able to determine the relative importance of clinical risk factors and identify patients at the highest risk of kidney function decline after hospitalization during the first wave of COVID-19 pandemic. In this study, we employed RF to identify the most significant factors associated with post-hospitalization eGFR decline in patients admitted at Stony Brook University Hospital (SBUH), and then used LR and CART to determine the relative importance of these factors and the patient sub-groups at the highest and lowest risk.

## Materials and method

### Study design and participants

We conducted a retrospective cohort study on patients hospitalized at Stony Brook University Hospital (SBUH) from March 6th, 2020, to December 31st, 2020, during the first year of the COVID-19 pandemic. SBUH is the largest tertiary care centers in Long Island, NY that treated one of the highest numbers of COVID-19 patients in the United States in 2020. All patients who were discharged alive from the hospital were followed until February 16th, 2023, for assessment of kidney function (using outpatient eGFR values). We excluded patients who were ≤18 years of age, pregnant, or who had end stage kidney disease (ESKD), including chronic dialysis or kidney transplant. The study was approved by the SBU Institutional Review Board (IRB # 2020–00239).

Consent was not obtained (waived) since the data were deidentified and analyzed anonymously.

### Privacy protection and data security

The data were accessed for research purposes on 01/06/2023 and 01/12/2023. Initially, the authors involved in the formal analysis had access to Protected Health Information (PHI) when it was extracted from the SBUH electronic health records (EHR). These files were stored on a secure, HIPAA-compliant server. PHI was promptly removed, and all subsequent analyses were conducted on de-identified patient data. All other authors had access only to de-identified data and summary results during research meetings.

### Outcome (eGFR decline) definition

Hospital survivors were followed for 36 months post discharge from index hospitalization. For each patient, a baseline eGFR was identified during or before the index hospitalization. The change in eGFR was estimated using this baseline eGFR and the most recent eGFR for all patients with an outpatient eGFR value 90 or more days after discharge from the index hospitalization. The target outcome of "fast eGFR decline" was defined as $\geq 5$ ml/min/1.73 m$^2$ per year [32]. The 'control group' was patients without fast eGFR decline during follow-up.

### Data collection and definition of other variables

Information pertaining to data collection and the definition of variables is detailed in the **S1 Methods** section.

### Univariate analyses

If more than 5% of values for a variable were missing, the variable was removed from the analysis. In each subset and propensity matched dataset, univariate LR was used to identify potential risk factors for the target outcome (fast versus not fast eGFR decline). For continuous variables, we summarized the mean value and standard deviation for each patient group (ICU admitted *versus* not admitted, COVID-19 positive *versus* COVID-19 negative, and with fast eGFR decline *versus* without fast eGFR decline). For binary variables (true or false), we determined the number and corresponding proportion in each patient group. P-value $< 0.05$ was considered as the cutoff for statistical significance.

### Summary of the multivariate and machine learning analyses

We generated multivariate LR models and CART decision trees to predict whether hospital survivors will develop fast eGFR decline during the follow-up period. The potential risk factors were selected based on RF analyses. By default, in each RF model, the number of decision trees was set to 500 and the number of variables used as a potential candidate split variable was 3. We chose the union of the top 10 variables based on the mean decrease accuracy (MDA) and the mean decrease in Gini (MDG) coefficient to fit LR and CART models.

In multivariate LR models, variables with significant odds ratio were considered as influential factors. In CART decision trees analyses, the minimum number of observations allowed in a terminal node was set to be 2% of the sample size N. The maximum depth of the decision tree was set to be 3. Variables that appeared as the split condition for the nodes in the decision tree were considered influential factors. We also calculated the overall prediction accuracy, sensitivity and specificity of each LR model and CART decision tree to compare the model performances.

The details of LR, RF and CART methods are in the **S1 Methods** section.

## Propensity score matching (PSM) analysis

PSM is a quasi-experimental method used to establish an artificial control group by matching characteristics of each study unit with a control unit to better estimate the impact of a predictor. In our study, PSM was conducted using nearest neighbor matching in a 1:1 ratio without replacement. Propensity scores were estimated through logistic regression based on the important demographic covariates: sex, race, ethnicity and age. This method ensures that each treated unit (ICU admission or COVID-19 positive) is matched to a control unit (non-ICU admission or COVID-19 negative respectively) with the closest propensity score, thereby minimizing group differences and enabling direct comparisons.

To assess balance, we calculated absolute standardized mean differences (SMDs) for both the unadjusted (pre-matching) and adjusted (post-matching) datasets. The absolute mean difference plots demonstrate that all SMDs post-matching decreased to below 0.10, which is a commonly used threshold for acceptable balance, as previously reported [33]. This substantial reduction in SMDs indicates a significant improvement in balance after matching, ensuring the comparability of the matched groups (S13 and S14 Figs).

## Statistical analysis

All statistical analyses and machine learning analyses were performed using R 4.2.3.

## Results

## 1. Comparison of patients with and without fast eGFR decline

Of the cohort of 1,747 hospital survivors, 61.6% were noted to have a fast eGFR decline during a mean follow-up of 214.33 (±109.28) days (**Table 1**). Patients in the fast eGFR decline sub-group had a mean eGFR decline of 35.04 ml/min/1.73 m$^2$ per year (standard deviation [SD] 34.44), whereas those in the control group (without fast eGFR decline) had a mean rise in eGFR of 11.84 ml/min/1.73 m$^2$ per year (SD 21.00). Patients with fast eGFR decline were more likely to be younger, have a higher baseline eGFR, and had greater severity of hospital illness as reflected by a greater length of hospital stay (LOHS) and increased likelihood of requiring intensive care unit (ICU) admission, mechanical ventilation (MV), and vasopressors (**Table 1**). Patients with fast eGFR decline were 1.4 times more likely to have moderate/severe AKI (stages 2 and 3 - AKI-2/3) during hospitalization (11.9 vs 8.4%).

Patients with fast eGFR decline were more likely to have a higher baseline eGFR (90.51 ± 29.52 mL/min/1.73m$^2$) compared to those without fast eGFR decline (78.91 ± 29.12 mL/min/1.73m$^2$). On further categorization by baseline eGFR, those with baseline eGFR >120 mL/min/1.73m$^2$ were significantly more likely to be fast eGFR decliners, while those with baseline eGFR <60 mL/min/1.73m$^2$ were more likely to be in the group without fast eGFR decline (**Table 1**).

**1a. Machine learning analysis.** In <u>RF analysis</u>, LOHS was among the top three variables associated with fast versus not fast eGFR decline in both the MDA and MDG plots (**S1 Fig**). The other top variables included vasopressor use, age, COPD and BMI. Also, among the top 3 variables were AKI-2/3 in the *COVID-19 negative* subset (**S2 Fig**) and Hispanic ethnicity, MV and MV days in the *COVID-19 positive* subset (**S3 Fig**). The top ten variables from RF analysis were used for LR and CART decision-tree analysis.

In <u>multivariate LR analysis</u> of the whole cohort, longer LOHS, and vasopressor use were significantly associated with greater odds of fast (vs. not fast) eGFR decline whereas older age was associated with lower odds (**Table 2**). Greater odds of fast (vs. not fast) eGFR decline were

**Table 1. Univariate analysis of the followed-up patients with and without fast eGFR decline.**

| Variables | Total | N = 1747 | Without Fast eGFR decline | N = 671 38.41% | Fast eGFR decline | N = 1076 61.59% | |
|---|---|---|---|---|---|---|---|
| N = 1747 | (Mean/N) | (Std/%) | (Mean/N) | (Std/%) | (Mean/N) | (Std/%) | P-value |
| **Demographics** | | | | | | | |
| Sex (N, %) | | | | | | | |
| Male | 962 | 55.07% | 362 | 53.95% | 600 | 55.76% | 0.459 |
| Female | 785 | 44.93% | 309 | 46.05% | 476 | 44.24% | 0.459 |
| Race (N, %) | | | | | | | |
| White | 1270 | 72.70% | 491 | 73.17% | 779 | 72.40% | 0.723 |
| Non-White | 477 | 27.30% | 180 | 26.83% | 297 | 27.60% | 0.723 |
| Unknown | 323 | 18.49% | 125 | 18.63% | 198 | 18.40% | 0.905 |
| Ethnicity (N, %) | | | | | | | |
| Non-Hispanic | 1298 | 74.30% | 489 | 72.88% | 809 | 75.19% | 0.283 |
| Hispanic | 189 | 10.82% | 73 | 10.88% | 116 | 10.78% | 0.949 |
| Unknown | 260 | 14.88% | 109 | 16.24% | 151 | 14.03% | 0.207 |
| Age (Mean, SD) | 63.81 | 17.95 | 65.23 | 17.31 | 62.92 | 18.29 | **0.009** |
| **Co-morbid conditions (N, %)** | | | | | | | |
| DM | 548 | 31.37% | 206 | 30.70% | 342 | 31.78% | 0.635 |
| HF | 404 | 23.13% | 146 | 21.76% | 258 | 23.98% | 0.285 |
| CKD | 396 | 22.67% | 156 | 23.25% | 240 | 22.30% | 0.647 |
| COPD | 221 | 12.65% | 88 | 13.11% | 133 | 12.36% | 0.645 |
| HTN | 894 | 51.17% | 355 | 52.91% | 539 | 50.09% | 0.253 |
| CAD | 543 | 31.08% | 211 | 31.45% | 332 | 30.86% | 0.795 |
| Cancer | 362 | 20.72% | 141 | 21.01% | 221 | 20.54% | 0.812 |
| Asthma | 139 | 7.96% | 55 | 8.20% | 84 | 7.81% | 0.770 |
| Psychiatric diagnosis | 968 | 55.41% | 358 | 53.35% | 610 | 56.69% | 0.172 |
| BMI (Mean, SD) | 28.62 | 8.47 | 28.53 | 7.54 | 28.67 | 9.01 | 0.751 |
| **Severity of illness** | | | | | | | |
| LOHS (Mean, SD) | 8.44 | 11.22 | 6.62 | 7.62 | 9.58 | 12.84 | **<0.001** |
| ICU admission (N, %) | 306 | 17.52% | 83 | 12.37% | 223 | 20.72% | **<0.001** |
| MV (N, %) | 89 | 5.09% | 17 | 2.53% | 72 | 6.69% | **<0.001** |
| MV days (Mean, SD) | 0.61 | 4.36 | 0.14 | 1.01 | 0.90 | 5.48 | **0.002** |
| ARDS (N, %) | 20 | 1.14% | 4 | 0.60% | 16 | 1.49% | 0.100 |
| Vasopressor (N, %) | 400 | 22.90% | 116 | 17.29% | 284 | 26.39% | **<0.001** |
| Sepsis (N, %) | 244 | 13.97% | 80 | 11.92% | 164 | 15.24% | 0.052 |
| **AKI_23** | 184 | 10.53% | 56 | 8.35% | 128 | 11.90% | **0.019** |
| **COVID-19** | 260 | 14.88% | 92 | 13.71% | 168 | 15.61% | 0.278 |
| **Kidney function measures** | | | | | | | |
| Baseline eGFR | 86.06 | 29.90 | 78.91 | 29.12 | 90.51 | 29.52 | **<0.001** |
| Baseline eGFR > 120 | 189 | 10.82% | 34 | 5.07% | 155 | 14.41% | **<0.001** |
| Baseline eGFR 90 to 120 | 693 | 39.57% | 254 | 37.85% | 439 | 40.80% | 0.241 |
| Baseline eGFR 60 to 89 | 494 | 28.28% | 204 | 30.40% | 290 | 26.96% | 0.133 |
| Baseline eGFR 30 to 59 | 292 | 16.71% | 134 | 19.97% | 158 | 14.68% | **0.005** |
| Baseline eGFR 15 to 29 | 62 | 3.55% | 32 | 4.77% | 30 | 2.79% | **0.041** |
| Baseline eGFR < 15 | 17 | 0.97% | 13 | 1.94% | 4 | 0.37% | **0.003** |
| Final eGFR | 77.38 | 30.11 | 84.28 | 27.72 | 73.07 | 30.75 | **<0.001** |
| Change in eGFR | -8.68 | 17.10 | 5.37 | 9.26 | -17.45 | 14.87 | **<0.001** |
| Follow-up days | 214.33 | 109.28 | 226.06 | 124.70 | 207.02 | 97.80 | **0.001** |
| eGFR change per year | -17.03 | 37.68 | 11.84 | 21.00 | -35.04 | 34.44 | 0.552 |

*(Continued)*

**Table 1.** (Continued)

| Variables | Total | N = 1747 | Without Fast eGFR decline | N = 671 38.41% | Fast eGFR decline | N = 1076 61.59% | |
|---|---|---|---|---|---|---|---|
| N = 1747 | (Mean/N) | (Std/%) | (Mean/N) | (Std/%) | (Mean/N) | (Std/%) | P-value |
| Other lab measures | | | | | | | |
| WBC | 8.03 | 4.19 | 8.00 | 5.13 | 8.05 | 3.49 | 0.813 |
| Hb | 11.37 | 2.07 | 11.49 | 2.05 | 11.29 | 2.08 | **0.049** |
| Platelets | 251.65 | 122.42 | 242.97 | 124.08 | 257.04 | 121.13 | **0.020** |

Categorical variables presented as a count with associated percentage, continuous variables presented as value with standard deviation (Std). Univariate logistic p-values < 0.05 were considered significant and have been bolded.

Abbreviations: DM = diabetes mellitus, HF = heart failure, CKD = chronic kidney disease, COPD = chronic obstructive pulmonary disease, HTN = hypertension, CAD = coronary artery disease, BMI = Body Mass Index, LOHS = length of hospital stay, ICU admission = intensive care unit admission, MV = mechanical ventilation, ARDS = acute respiratory distress syndrome, AKI = acute kidney injury, COVID-19 = Corona virus disease 2019, eGFR = estimated glomerular filtration rate (mL/min/1.73m$^2$), WBC = White Blood Cell count, Hb = Hemoglobin, Platelets = Platelet count.

significantly associated with longer LOHS and vasopressor use in the *COVID-19 negative* subset (**S1 Table**) and with diabetes mellitus (DM) in the *COVID-19 positive* subgroup (**S2 Table**).

In CART decision tree analysis of the full cohort, LOHS was the most important factor followed by vasopressor use and AKI-2/3 diagnosis (**Fig 1**). Those with AKI and hospital stay ≥ 27 days had the highest likelihood of fast (vs. not fast) eGFR decline, whereas the lowest risk was found in two subgroups (those without vasopressor use and body mass index [BMI] ≥ 22 and those with vasopressor use, but with a hospital stay < 6 days) (**Fig 1**). In the *COVID-*

**Table 2. Logistic regression for fast eGFR decline in the original dataset (N = 1747).**

| Variable | | OR (univariable) | OR (multivariable) |
|---|---|---|---|
| LOHS | Mean (SD) | 1.03 (1.02–1.05, ***) | **1.02 (1.01–1.04, **)** |
| Vasopressor | 1 | 1.72 (1.35–2.19, ***) | **1.45 (1.12–1.89, **)** |
| COPD | 1 | 0.93 (0.70–1.25) | 0.95 (0.70–1.29) |
| Age | Mean (SD) | 0.99 (0.99–1.00, **) | **0.99 (0.99–1.00, *)** |
| MV days | Mean (SD) | 1.13 (1.06–1.23, **) | 1.05 (0.98–1.18) |
| Asthma | 1 | 0.95 (0.67–1.36) | 0.92 (0.64–1.34) |
| White | 1 | 0.96 (0.77–1.19) | 1.03 (0.82–1.30) |
| AKI_23 | 1 | 1.48 (1.07–2.08, *) | 0.90 (0.62–1.31) |
| MV | 1 | 2.76 (1.65–4.87, ***) | 1.03 (0.45–2.26) |
| CKD | 1 | 0.95 (0.75–1.19) | 0.97 (0.76–1.26) |
| BMI | Mean (SD) | 1.00 (0.99–1.01) | 1.00 (0.99–1.01) |
| Male | 1 | 1.08 (0.89–1.31) | 1.01 (0.83–1.24) |
| Psychiatric diagnosis | 1 | 1.14 (0.94–1.39) | 1.09 (0.89–1.33) |
| HTN | 1 | 0.89 (0.74–1.08) | 0.91 (0.74–1.12) |
| DM | 1 | 1.05 (0.85–1.30) | 1.09 (0.87–1.36) |
| CAD | 1 | 0.97 (0.79–1.20) | 1.08 (0.86–1.37) |
| Cancer | 1 | 0.97 (0.77–1.23) | 0.95 (0.74–1.21) |

The top variables form Random Forest analysis were selected for Logistic Regression analysis.

P-values < 0.05 were considered significant and were summarized with '*', p-values < 0.01 were considered significant and were summarized with '**', and p-values < 0.001 were considered significant and were summarized with '***'.

Abbreviations: LOHS = length of hospital stay, COPD = chronic obstructive pulmonary disease, MV = mechanical ventilation, CKD = chronic kidney disease, HTN = hypertension, DM = diabetes mellitus, CAD = coronary artery disease, eGFR = estimated glomerular filtration rate.

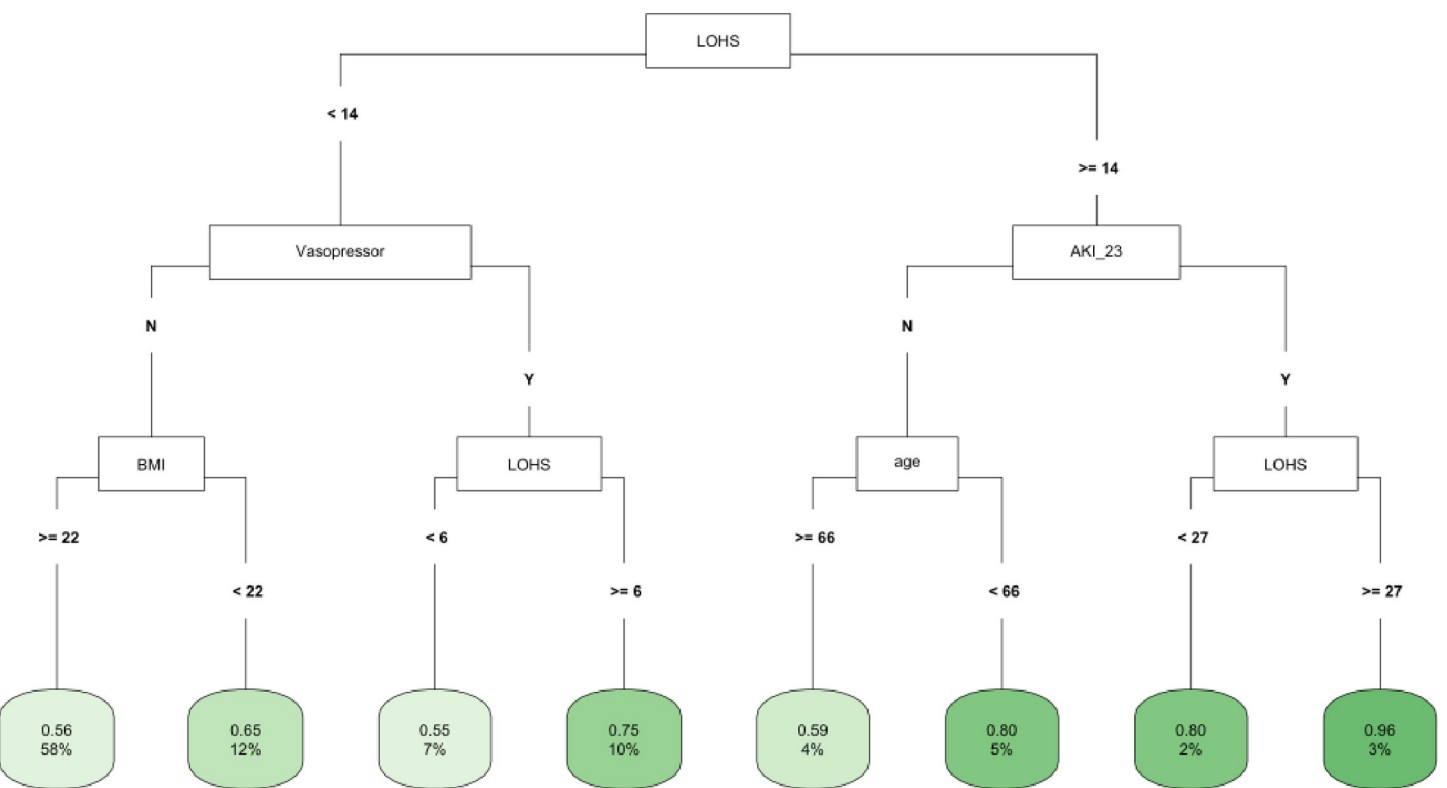

**Fig 1. CART decision tree for fast eGFR decline in the overall cohort.** The number of observations in a terminal node was set as at least 2% of the sample size. The percentage mentioned in the terminal node is the % of patients of the starting cohort of the analyses. In each terminal node, the risk of fast eGFR decline (vs. not fast) ranges from 0.00 (lowest) to 1.00 (highest). The color of the terminal node represents the risk associated with the tree attached to each node, with the intensity of green color indicating a stronger risk, while intensity of blue color representing a lower risk. The maximum depth of the decision tree was set to be 3.

*19 negative* subgroup, LOHS was again the most important risk factor followed by vasopressor use (**Fig 2**). Patients with hospital stay ≥ 6 days, with vasopressor use and of female sex had the highest likelihood of fast eGFR decline, whereas those with hospital stay < 2 days and with coronary artery disease (CAD) had the lowest risk (**Fig 2**). In the *COVID-19 positive* subgroup, LOHS was also the most important risk factor followed by cancer diagnosis and age (**Fig 3**). Patients with hospital stay ≥ 15 days, age < 65 and BMI < 27 had the highest likelihood of fast eGFR decline, whereas those with hospital stay < 15 days with cancer and without hypertension (HTN) had the lowest risk (**Fig 3**).

## 2. Comparison of patients with and without ICU admission after PSM for demographics

Patients admitted to the ICU were more likely to require vasopressors or have a greater LOHS or diagnosis of sepsis (**S3 Table**). Patients with ICU admission were 3.7 times more likely to have AKI-2/3 during hospitalization and had a greater proportion of patients with fast eGFR decline (72.9 vs 53.9%) during follow-up compared those not admitted to the ICU.

**2a. Machine learning analysis.** In the PSM matched cohort based on ICU status, LOHS was again among the top three variables associated with fast (vs. not fast) eGFR decline in RF analysis in both the MDA and MDG plots (**S4 Fig**). The other top variables were MV days, vasopressor use, age, and BMI. The top three variables also included ICU admission among

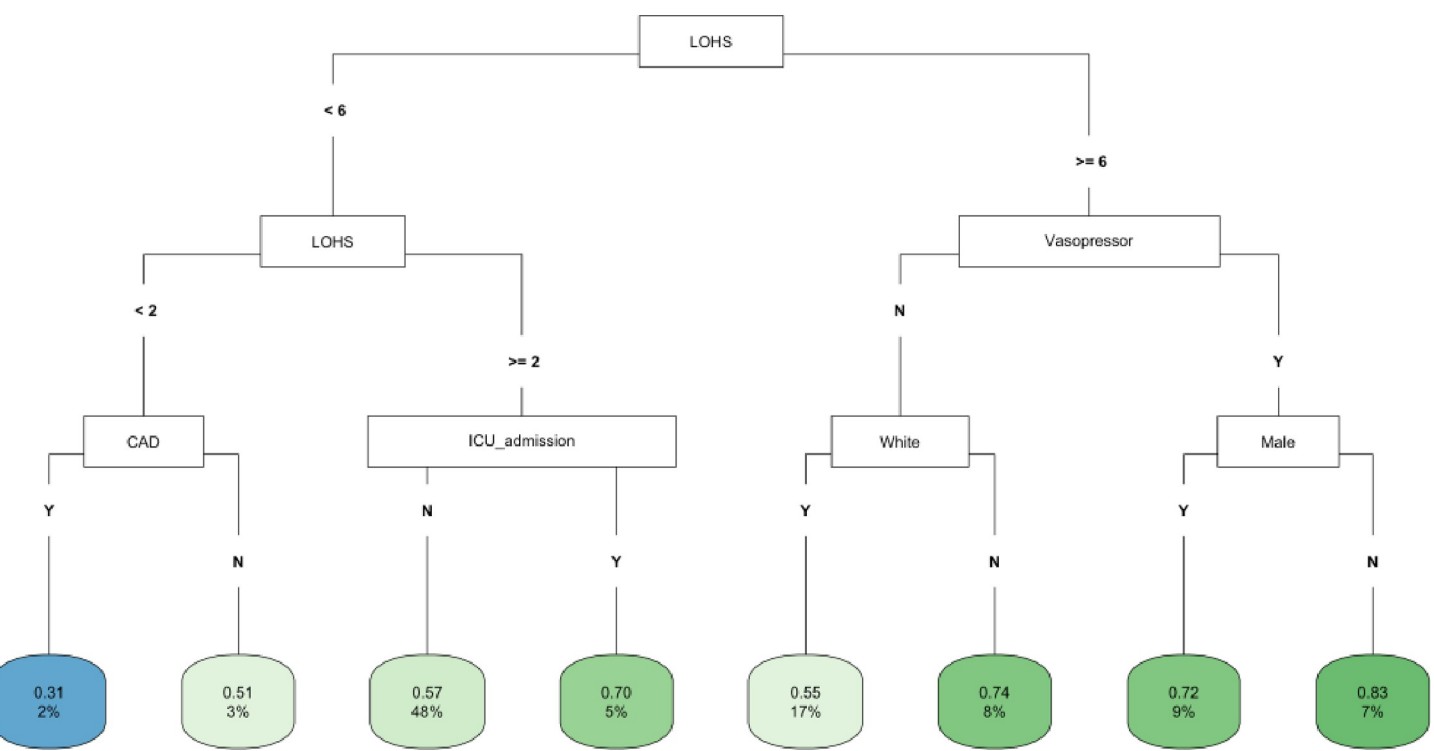

**Fig 2. CART decision tree for fast eGFR decline in the COVID negative subset.** The number of observations in a terminal node was set as at least 2% of the sample size. The percentage mentioned in the terminal node is the % of patients of the starting cohort of the analyses. In each terminal node, the risk of fast eGFR decline (vs. not fast) ranges from 0.00 (lowest) to 1.00 (highest). The color of the terminal node represents the risk associated with the tree attached to each node, with the intensity of green color indicating a stronger risk, while intensity of blue color representing a lower risk. The maximum depth of the decision tree was set to be 3.

the *COVID-19 negative* subgroup (**S5 Fig**) and MV among the *COVID-19 positive* subgroup (**S6 Fig**).

In <u>multivariate LR analysis,</u> longer LOHS and ICU admission were significantly associated with greater odds of fast (vs. not fast) eGFR decline whereas baseline CKD was associated with lower odds (**Table 3**). In the *COVID-19 negative* subset (**S4 Table**), older age and male sex were significantly associated with lower odds, whereas in the *COVID-19 positive* subgroup (**S5 Table**), DM was associated with greater odds of fast eGFR decline.

In <u>CART analysis</u>, ICU admission was the most important factor followed by LOHS and baseline CKD (**Fig 4**). The subgroups with ICU admission and either LOHS $\geq$ 32 days and White race, or LOHS< 32 days and age < 32 years had the highest likelihood of fast eGFR decline. The lowest risk occurred in those without an ICU admission who had CKD at baseline and a BMI<24 (**Fig 4**). In the *COVID-19 negative* subgroup, age was the most important factor followed by ICU admission and LOHS (**S7 Fig**). Patients with age < 70 years and hospital stay $\geq$ 44 days had the highest likelihood of fast eGFR decline, whereas those with age $\geq$ 70 years but < 81 years and no ICU admission had the lowest risk (**S7 Fig**). In the *COVID-19 positive* subgroup, LOHS was again the most important factor followed by DM diagnosis and BMI (**S8 Fig**). Patients with hospital stay $\geq$ 24 days, BMI < 35 and age $\geq$ 37 years had the highest likelihood of fast eGFR decline, whereas those with DM diagnosis but hospital stay < 3 days had the lowest risk (**S8 Fig**).

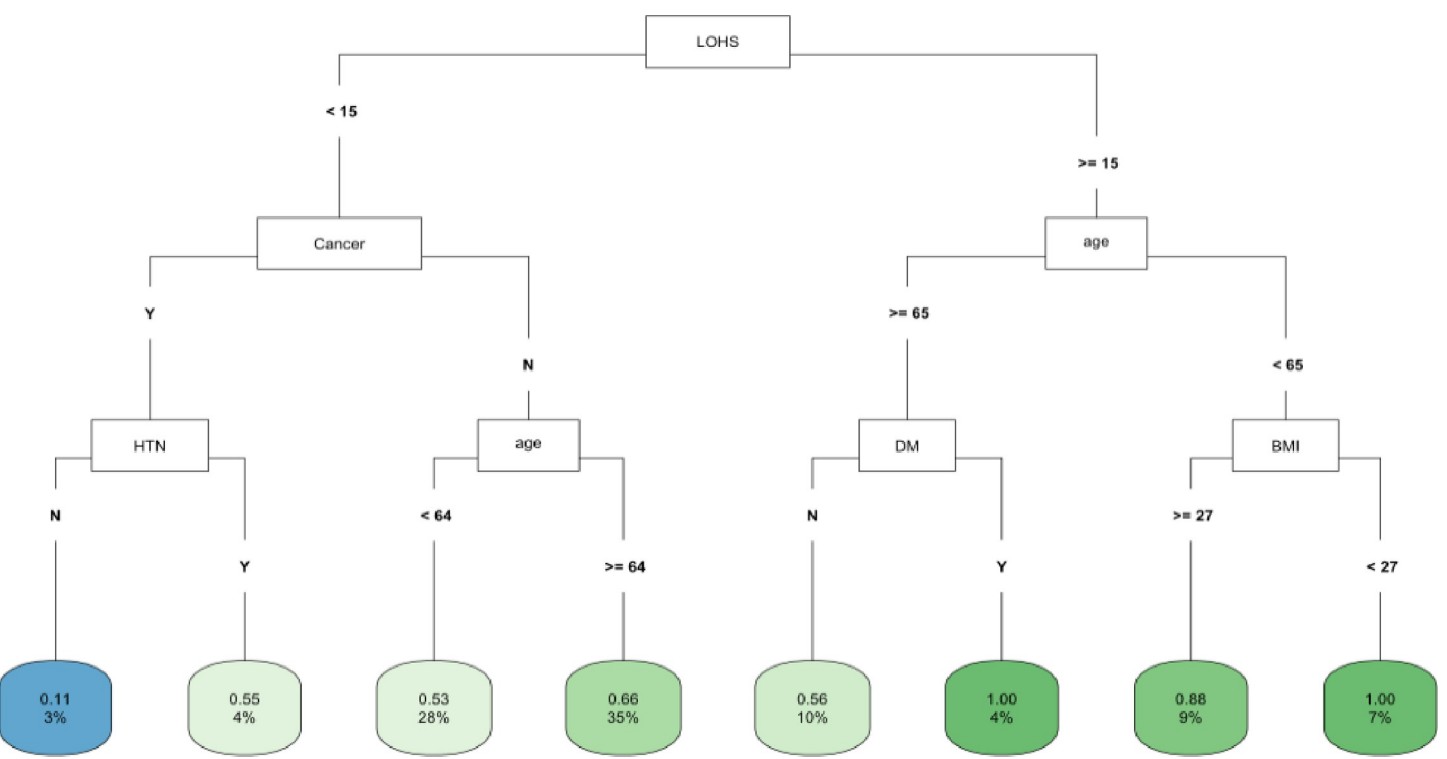

**Fig 3. CART decision tree for fast eGFR decline in the COVID positive subset.** The number of observations in a terminal node was set as at least 2% of the sample size. The percentage mentioned in the terminal node is the % of patients of the starting cohort of the analyses. In each terminal node, the risk of fast eGFR decline (vs. not fast) ranges from 0.00 (lowest) to 1.00 (highest). The color of the terminal node represents the risk associated with the tree attached to each node, with the intensity of green color indicating a stronger risk, while intensity of blue color representing a lower risk. The maximum depth of the decision tree was set to be 3.

### 3. Comparison of patients with and without COVID-19 diagnosis after PSM for demographics

In the overall cohort, compared to those without COVID-19, patients admitted with COVID-19 diagnosis were more likely to have a greater mean decline in eGFR during follow-up, but the proportion of patients with fast eGFR decline was not significantly different (**S6 Table**). There was no significant difference in baseline age, sex or eGFR between the COVID positive and negative groups.

In the sub-group of patients who were propensity matched for demographics, compared to those without COVID-19, patients admitted with COVID-19 diagnosis were less likely to require vasopressors or have baseline HTN, CAD, and cancer; but were more likely to have an ICU admission, sepsis, ARDS, MV, and a greater LOHS (**S7 Table**). Patients with COVID-19 were 1.9 times more likely to have AKI-2/3 during hospitalization and a greater mean decline in eGFR during follow-up, but the proportion of patients with fast eGFR decline was not significantly different.

**3a. Machine learning analysis.** In the PSM matched cohort based on COVID status, LOHS was again among the top 3 variables associated with fast (vs not fast) eGFR decline in RF analysis in both the MDA and MDG plots (**S9 Fig**). The other top variables were MV days, White race, age and BMI. The top 3 variables also included vasopressor use and Hispanic ethnicity among the *COVID-19 negative* subgroup (**S10 Fig**) and MV and Hispanic ethnicity among *COVID-19 positive* subset (**S11 Fig**).

**Table 3. Logistic regression for fast eGFR decline in the PSM matched ICU subset of the whole cohort (N = 612).**

| Variable | | OR (univariable) | OR (multivariable) |
|---|---|---|---|
| LOHS | Mean (SD) | 1.04 (1.02–1.06, ***) | **1.03 (1.00–1.05, *)** |
| MV days | Mean (SD) | 1.13 (1.05–1.23, **) | 1.04 (0.97–1.19) |
| Vasopressor | 1 | 1.89 (1.33–2.70, ***) | 1.17 (0.75–1.81) |
| CKD | 1 | 0.60 (0.40–0.88, **) | **0.61 (0.39–0.95, *)** |
| ICU admission | 1 | 2.30 (1.64–3.23, ***) | **1.71 (1.14–2.59, **)** |
| COVID | 1 | 1.05 (0.67–1.66) | 0.76 (0.46–1.27) |
| Cancer | 1 | 0.91 (0.61–1.38) | 0.82 (0.53–1.28) |
| AKI_23 | 1 | 1.82 (1.15–2.94, *) | 0.89 (0.51–1.58) |
| COPD | 1 | 0.88 (0.54–1.44) | 0.93 (0.55–1.56) |
| MV | 1 | 2.77 (1.63–4.99, ***) | 0.91 (0.37–2.13) |
| BMI | Mean (SD) | 1.00 (0.98–1.02) | 0.99 (0.97–1.02) |
| Age | Mean (SD) | 0.99 (0.98–1.00, *) | 0.99 (0.98–1.00) |
| Psychiatric diagnosis | 1 | 1.21 (0.87–1.68) | 0.96 (0.67–1.37) |
| HTN | 1 | 0.91 (0.66–1.27) | 0.82 (0.57–1.19) |
| Male | 1 | 0.86 (0.61–1.21) | 0.86 (0.60–1.23) |
| DM | 1 | 0.98 (0.69–1.40) | 1.14 (0.78–1.69) |

The top variables form Random Forest analysis were selected for Logistic Regression analysis.

P-values < 0.05 were considered significant and were summarized with '*', p-values < 0.01 were considered significant and were summarized with '**', and p-values < 0.001 were considered significant and were summarized with '***'.

Abbreviations: LOHS = length of hospital stay, COPD = chronic obstructive pulmonary disease, MV = mechanical ventilation, CKD = chronic kidney disease, HTN = hypertension, DM = diabetes mellitus, CAD = coronary artery disease. eGFR = estimated glomerular filtration rate.

In <u>multivariate LR analysis</u>, DM was significantly associated with greater odds of fast (vs. not fast) eGFR decline whereas older age was associated with lower odds (**Table 4**). In the *COVID-19 negative* subset, there were no statistically significant variables in multivariate LR (**S8 Table**). The findings of the *COVID-19 positive* subgroup are noted in **S2 Table**.

In <u>CART analysis</u>, LOHS was the most important factor followed by BMI and age (**Fig 5**). Those with hospital stay ≥ 8 days and age < 46 but ≥36 years had the highest likelihood of fast (vs. not fast) eGFR decline, whereas those with hospital stay < 8 days, BMI ≥37 but no DM diagnosis had the lowest risk (**Fig 5**). In the *COVID-19 negative* subgroup, LOHS was again the most important factor followed by BMI (**S12 Fig**). Patients with hospital stay ≥ 8 but < 12 days had the highest likelihood of fast) eGFR decline, whereas those with hospital stay < 2 days and BMI ≥ 31 had the lowest risk. The findings of the COVID-19 positive subgroup are noted in **Fig 3**.

## 4. Comparison of LR and CART methods

Both CART and LR methods showed similar accuracy and predictive power (**S9 Table**) in each dataset (primary and PSM cohorts). Both methods showed high specificity but low sensitivity for outcome prediction. The sensitivity and accuracy were comparatively higher in patients diagnosed with COVID-19 compared to those without.

## Discussion

In this study of 1,747 hospital survivors with and without COVID-19, we identified age, baseline eGFR, greater severity of hospital illness, and moderate/severe AKI as the key hospital factors associated with fast kidney function decline after hospitalization. Among the factors

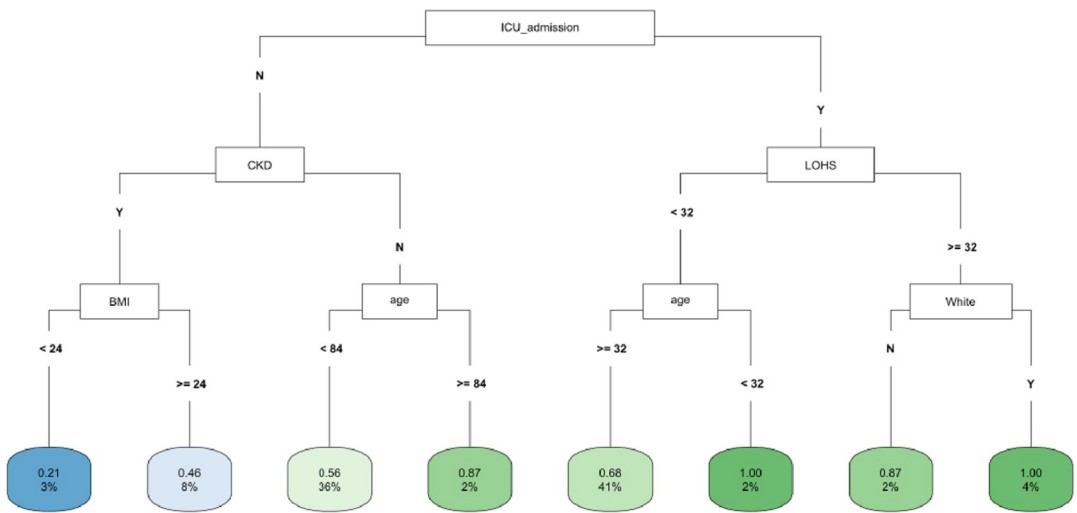

**Fig 4. CART decision tree for fast eGFR decline in the PSM matched ICU subset of the whole cohort.** The number of observations in a terminal node was set as at least 2% of the sample size. The percentage mentioned in the terminal node is the % of patients of the starting cohort of the analyses. In each terminal node, the risk of fast eGFR decline (vs. not fast) ranges from 0.00 (lowest) to 1.00 (highest). The color of the terminal node represents the risk associated with the tree attached to each node, with the intensity of green color indicating a stronger risk, while intensity of blue color representing a lower risk. The maximum depth of the decision tree was set to be 3.

associated with severity of hospital illness, length of hospital stay (LOHS) was the most important factor followed by vasopressor use in both LR and CART analysis. Using CART, we were able to identify the patient sub-groups with the highest and lowest risk of post-hospitalization fast eGFR decline. After stratification of the cohort based on ICU and COVID-19 status, other risk factors such as admission to the ICU and baseline CKD, DM, and BMI were identified. In

**Table 4. Logistic regression for fast eGFR decline in the PSM matched COVID-19 subset of the whole cohort (N = 520).**

| Variable | | OR (univariable) | OR (multivariable) |
|---|---|---|---|
| LOHS | Mean (SD) | 1.04 (1.02–1.06, ***) | 1.02 (1.00–1.05) |
| White | 1 | 1.17 (0.80–1.69) | 1.30 (0.85–2.00) |
| MV days | Mean (SD) | 1.11 (1.03–1.24, *) | 1.03 (0.96–1.16) |
| Hispanic | 1 | 0.77 (0.51–1.18) | 0.69 (0.42–1.14) |
| Vasopressor | 1 | 2.15 (1.35–3.51, **) | 1.54 (0.91–2.66) |
| Age | Mean (SD) | 0.99 (0.98–1.00) | **0.99 (0.97–1.00, *)** |
| HTN | 1 | 1.45 (1.01–2.07, *) | 1.45 (0.99–2.14) |
| Sepsis | 1 | 1.78 (1.11–2.93, *) | 1.39 (0.83–2.36) |
| DM | 1 | 1.31 (0.89–1.93) | **1.55 (1.02–2.38, *)** |
| COVID | 1 | 1.18 (0.83–1.68) | 1.07 (0.71–1.61) |
| BMI | Mean (SD) | 0.99 (0.97–1.01) | 0.98 (0.95–1.00) |
| Male | 1 | 0.97 (0.68–1.39) | 0.81 (0.55–1.17) |
| Psychiatric diagnosis | 1 | 1.13 (0.79–1.61) | 0.96 (0.65–1.41) |

The top variables form Random Forest analysis were selected for Logistic Regression analysis.

P-values < 0.05 were considered significant and were summarized with '*', p-values < 0.01 were considered significant and were summarized with '**', and p-values < 0.001 were considered significant and were summarized with '***'.

Abbreviations: LOHS = length of hospital stay, COPD = chronic obstructive pulmonary disease, MV = mechanical ventilation, CKD = chronic kidney disease, HTN = hypertension, DM = diabetes mellitus, CAD = coronary artery disease, eGFR = estimated glomerular filtration rate.

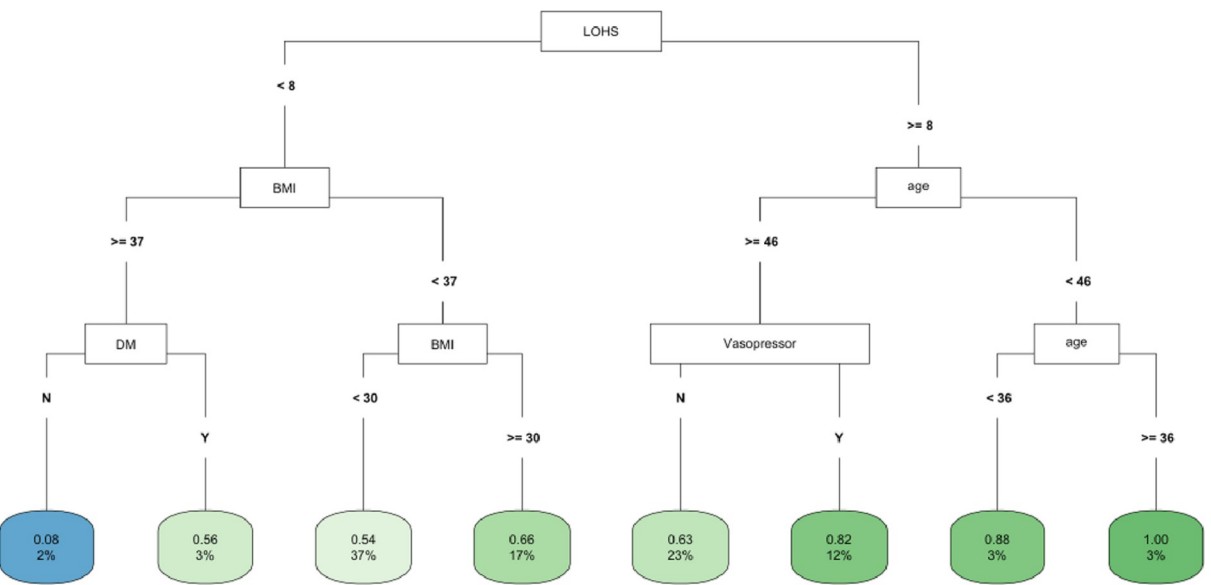

**Fig 5. CART decision tree for fast eGFR decline in the PSM matched COVID-19 subset of the whole cohort.** The number of observations in a terminal node was set as at least 2% of the sample size. The percentage mentioned in the terminal node is the % of patients of the starting cohort of the analyses. In each terminal node, the risk of fast eGFR decline (vs. not fast) ranges from 0.00 (lowest) to 1.00 (highest). The color of the terminal node represents the risk associated with the tree attached to each node, with the intensity of green color indicating a stronger risk, while intensity of blue color representing a lower risk. The maximum depth of the decision tree was set to be 3.

all analyses, LOHS emerged as a highly significant factor. To our knowledge this is the first study to report the use of CART for identifying the patient sub-groups associated with the highest risk of fast post-hospitalization eGFR decline among patients hospitalized during the first year of the pandemic.

Severity of acute illness, patient characteristics, and complications during the hospitalization are associated with longer LOHS [34–37]. Longer LOHS has been associated with severity of kidney disease in the hospital [38, 39], and adverse post-discharge health outcomes [40, 41]. Similar findings have also been reported during the COVID-19 pandemic [42–44]. In a study of patients with COVID-19 who had AKI in the hospital, post-discharge eGFR decline was associated with longer hospitalization [45]. In this study, we report that length of stay during hospitalization is significantly associated with post-discharge eGFR decline in both LR and CART analyses in one of the largest patient cohorts to-date from the pandemic era. This finding has significant implications for the outpatient kidney monitoring and risk stratification of patients who were recently hospitalized.

Although the severity of illness in the hospital, including the diagnosis of AKI, is well known to be associated with worse long-term kidney outcomes, in our study we report other less recognized associations with fast post-discharge eGFR decline. For example, we found patients with fast eGFR decline were more likely to have a higher baseline eGFR. Baseline eGFR is known to be associated with eGFR change after AKI [46], and with faster decline in eGFR over time [6, 47, 48]. Pathologic glomerular hyperfiltration resulting in higher eGFR values might be a contributing factor [49]. While did not find a difference in CKD diagnosis (Table 1) in the eGFR decline groups, it's not clear why patients with low baseline eGFR might have slower eGFR decline, although the number of patients with eGFR < 60 mL/min/1.73m$^2$ in our study was low (only 21% of the cohort). This interesting association of baseline eGFR with eGFR decline certainly needs further exploration in research studies.

Another interesting finding in our study was that younger age was associated with higher risk of fast post-discharge eGFR decline. This association has been previously reported [50] and might be related to inadequacy of current eGFR estimates in older individuals [51] and possibly to different pathophysiological mechanisms of CKD progression [52]. Both findings have important clinical implications in renal monitoring of patients after hospital discharge.

As an advancement over previous models using Cox proportional hazards [53], newer ML-based models including gradient boosting, regression splines and random forest have been developed to predict kidney disease [54–60]. Decision tree analysis can identify the patient sub-groups at lowest and highest risk of the eGFR decline through resulting cluster and hierarchy of decision nodes. CART has been used in the study of kidney diseases [14–19], however, the use of CART to predict eGFR decline has rarely been reported. A recent study used CART to identify factors associated with eGFR decline but was limited by a small sample and restricted to patients with partial nephrectomy [28]. To our knowledge, our study is the first to use CART to analyze factors associated with post-discharge eGFR decline in patients with and without COVID-19. Using CART analysis, we found that LOHS was the most important factor followed by vasopressor use and AKI-2/3 diagnosis. Those with AKI and hospital stay $\geq$ 27 days had the highest likelihood of fast (vs. not fast) eGFR decline, whereas the lowest risk was found in two subgroups: those with no vasopressor use and BMI$\geq$ 22 and those with vasopressor use, but with a hospital stay $<$ 6 days. These findings highlight the importance of evaluating recent hospitalization data of patients while monitoring kidney function in outpatient clinics.

Our study also highlights the importance of utilizing a non-parametric supervised learning algorithm like CART to identify high and low risk patient sub-groups, rather than relying solely on a traditional approach like LR which only identifies individual clinical factors. The application of CART models in clinical practice, particularly in ICU settings, offers several advantages [61, 62] CART's ability to generate simple and interpretable decision trees makes it a practical tool for risk stratification and decision-making in high-risk environments. For ICU patients, who often experience a higher frequency of AKI due to severity illness and other nephrotoxic insults and consequently have a high risk of post-discharge eGFR decline and CKD progression. CART can help identify high-risk individuals early, allowing for targeted interventions such as nephroprotective measures or closer monitoring. Furthermore, CART's flexibility in handling complex interactions between variables, such as baseline renal function, comorbidities, laboratory parameters and treatment modalities, is particularly relevant for ICU populations. Integrating CART models into electronic health record systems could facilitate real-time risk assessments, improving patient outcomes [63, 64]. However, successful implementation requires rigorous validation of the model in diverse ICU settings and the availability of high-quality, real-time data. This should be the focus of future validation and implementation studies.

During the pandemic period, multiple factors including COVID-19 infection were found to be associated with increased odds of rapid kidney function decline [65]. However, there have been only a few studies that report the use of ML to predict eGFR decline in hospital survivors from the first year of the pandemic when the treatment of this disease was evolving, and mass public vaccinations had not yet started [66]. Vaid et al. used several ML models for predicting dialysis requirement and death in patients hospitalized with COVID-19 and found an XGBoost model without imputation to have the highest accuracy [67]. We had previously reported the use of XGBoost to predict recovery after AKI in patients with and without COVID-19 [30]. The use of ML models, especially decision tree analysis, to predict post-hospitalization eGFR decline in COVID-19 survivors has been rarely reported. In a small study of 37 critically ill patients with COVID-19 during the first year of the pandemic, One Rule and decision trees methods were used to classify patients for risk of CKD and mortality [68].

However, this study was limited in design, sample size and focused only on in-hospital outcomes. In our study, among patients with COVID-19, baseline DM was significantly associated with greater odds of fast eGFR decline whereas older age was associated with lower odds in multivariate LR analysis. In CART analysis, LOHS was the most important factor followed by BMI and age. This data provides valuable insights into high-risk patients admitted with COVID-19 who require closer kidney monitoring after discharge.

Our study shows comparable accuracy between LR and CART as previously reported [69, 70]. Besides CART, other decision trees used in the prediction of kidney disease have had variable accuracy compared to other ML techniques [23–27]. CART's strengths lie in its simplicity and ease of interpretation due to its binary tree structure, unlike methods with multi-way splits. Its tree pruning technique, which grows a large tree and prunes it to optimal size, effectively prevents overfitting. As a non-parametric approach that is applicable to both classification and regression tasks, CART does not assume data distribution, making it particularly effective for modeling nonlinear relationships that parametric methods like LR may not handle well.

In our study, we used RF for selecting the top ten most significant features associated with fast eGFR decline. We then undertook LR and CART analysis to study the relative importance of these features. Previous studies have shown that the employment of ML for feature selection before decision tree analyses increases accuracy [26, 71].

Our study had several limitations. Due to our study requirements of at least two outpatient eGFR values more than 90 days after hospital discharge, only a third of the hospital survivors during the pandemic had data available to evaluate post-discharge eGFR decline. We did not have accurate urine output data and AKI in the hospital was diagnosed by the serum creatinine criteria only. A significant proportion of hospitalized patients did not have pre-hospitalization baseline eGFR available, and in those cases, the lowest serum creatinine during hospitalization was used to estimate baseline eGFR. Since most patients in our cohort had normal baseline kidney function (mean baseline eGFR of 86.06 ±29.90 ml/min/1.73m$^2$ and mean final eGFR of 77.38 ±30.11 ml/min/1.73m$^2$), we were not able to study other outcomes associated with CKD progression such as incident CKD, > 40% eGFR decline or incident ESKD. The lack of data on proteinuria, retrospective design, and use of data from a single center are additional limitations.

In conclusion, we report that use of combined ML techniques can provide a comprehensive understanding of the key factors associated with kidney outcomes. CART analysis can help identify the subgroups of hospitalized patients with the highest and lowest risk of post-discharge eGFR decline. In this study, CART identified length of hospitalization as the most important factor. Based on these findings, we advise clinicians to consider the length of hospitalization in their post-discharge monitoring of kidney function. We anticipate that identification of high-risk hospitalized patients through CART can significantly improve the post-discharge clinical management. Further studies in other healthcare systems are required to validate our findings.

## Supporting information

**S1 Fig. Random Forest for fast eGFR decline in the original dataset (N = 1747).**
(DOCX)

**S2 Fig. Random Forest for fast eGFR decline in the COVID negative subset (N = 1487).**
(DOCX)

**S3 Fig. Random Forest for fast eGFR decline in the COVID positive subset (N = 260).**
(DOCX)

**S4 Fig. Random Forest for fast eGFR decline in the ICU matched subset (N = 612).**
(DOCX)

**S5 Fig. Random Forest for fast eGFR decline in the ICU matched COVID negative subset (N = 510).**
(DOCX)

**S6 Fig. Random Forest for fast eGFR decline in the ICU matched COVID positive subset (N = 102).**
(DOCX)

**S7 Fig. CART decision tree for fast eGFR decline in the COVID-19 negative subgroup of the PSM matched ICU subset (N = 510).**
(DOCX)

**S8 Fig. CART decision tree for fast eGFR decline in the COVID-19 positive subgroup of the PSM matched ICU subset (N = 102).**
(DOCX)

**S9 Fig. Random Forest for fast eGFR decline in the COVID matched subset (N = 520).**
(DOCX)

**S10 Fig. Random Forest for fast eGFR decline in the COVID matched COVID negative subset (N = 260).**
(DOCX)

**S11 Fig. Random Forest for fast eGFR decline in the COVID matched COVID positive subset (N = 260).**
(DOCX)

**S12 Fig. CART decision tree for fast eGFR decline in the COVID negative subgroup of the PSM matched COVID-19 subset (N = 260).**
(DOCX)

**S13 Fig. Standardized Mean Differences (SMDs) Before and After Matching for ICU Admission Groups.**
(DOCX)

**S14 Fig. Standardized Mean Differences (SMDs) Before and After Matching for COVID-19 Diagnosis Groups.**
(DOCX)

**S1 Table. Logistic regression for fast eGFR decline in the COVID negative subset (N = 1487).**
(DOCX)

**S2 Table. Logistic regression for fast eGFR decline in the COVID positive subset (N = 260).**
(DOCX)

**S3 Table. Univariate analysis of the followed-up patients admitted and not admitted to the ICU after PSM on 4 demographic variables.**
(DOCX)

**S4 Table. Logistic regression for fast eGFR decline in the COVID negative subgroup of the PSM matched ICU subset (N = 510).**
(DOCX)

**S5 Table. Logistic regression for fast eGFR decline in the COVID positive subgroup of the PSM matched ICU subset (N = 102).**
(DOCX)

**S6 Table. Univariate analysis of the followed-up patients with and without COVID-19 (N = 1,747).**
(DOCX)

**S7 Table. Univariate analysis of the followed-up patients with and without COVID-19 after PSM on 4 demographic variables.**
(DOCX)

**S8 Table. Logistic regression for fast eGFR decline in the COVID-negative subgroup of the PSM matched COVID-19 subset (N = 260).**
(DOCX)

**S9 Table. Comparisons between CART decision tree and logistic regression models.**
(DOCX)

**S1 Methods.**
(DOCX)

## Author Contributions

**Conceptualization:** Farrukh M. Koraishy.

**Data curation:** Weihao Wang.

**Formal analysis:** Weihao Wang.

**Investigation:** Weihao Wang, Wei Zhu, Janos Hajagos, Laura Fochtmann, Farrukh M. Koraishy.

**Methodology:** Wei Zhu, Laura Fochtmann, Farrukh M. Koraishy.

**Project administration:** Farrukh M. Koraishy.

**Resources:** Wei Zhu, Janos Hajagos, Farrukh M. Koraishy.

**Supervision:** Farrukh M. Koraishy.

**Validation:** Weihao Wang, Farrukh M. Koraishy.

**Visualization:** Weihao Wang.

**Writing – original draft:** Weihao Wang, Farrukh M. Koraishy.

**Writing – review & editing:** Weihao Wang, Wei Zhu, Janos Hajagos, Laura Fochtmann, Farrukh M. Koraishy.

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
