## [Decision Letter · Decision Letter 0]

26 Nov 2024

PONE-D-24-49407Classification and Regression Trees Analysis Identifies Patients at High Risk for Kidney Function Decline Following HospitalizationPLOS ONE

Dear Dr. KORAISHY,

Thank you for submitting your manuscript to PLOS ONE. After careful consideration, we feel that it has merit but does not fully meet PLOS ONE’s publication criteria as it currently stands. Therefore, we invite you to submit a revised version of the manuscript that addresses the points raised during the review process.

We look forward to receiving your revised manuscript.

Kind regards,

Keiko Hosohata, Ph.D.

Academic Editor

PLOS ONE

3. Please include a caption for figure 6, 7, 8, 9, 10, 11, 12.

4. We note you have included a table to which you do not refer in the text of your manuscript. Please ensure that you refer to Table 2 in your text; if accepted, production will need this reference to link the reader to the Table.

Additional Editor Comments (if provided):

Reviewers' comments:

Reviewer's Responses to Questions

**Comments to the Author**

1. Is the manuscript technically sound, and do the data support the conclusions?

Reviewer #1: Yes

Reviewer #2: Yes

2. Has the statistical analysis been performed appropriately and rigorously? 

Reviewer #1: Yes

Reviewer #2: Yes

3. Have the authors made all data underlying the findings in their manuscript fully available?

Reviewer #1: Yes

Reviewer #2: Yes

4. Is the manuscript presented in an intelligible fashion and written in standard English?

Reviewer #1: Yes

Reviewer #2: Yes

5. Review Comments to the Author

Reviewer #1: Dear Authors

The current manuscript is outstanding but needs improvement in some areas. Even so, there are some questions.

Introduction

The authors should report the strong relationship between CKD and AKI, where one type of kidney injury can predispose to the other in both directions and both ways.

The authors should briefly report the frequency of AKI in hospitalized patients with COVID-19 and without COVID-19 to better contextualize.

Methods

The researchers should be able to classify kidney injury by the time of onset, such as AKI lasting up to 7 days, AKD between 7 and 30 days, and CKD lasting more than 30 days.

The authors analysed a lot of clinical data to understand the progression of CKD after COVID-19. However, it would be interesting to also analyse clinical laboratory data, such as complete blood count, to understand the relationship between hemoglobin concentration, leukocyte count, and platelet count with the progression to CKD.

Results

Was Machine Learning analysis able to observe other clinical factors related to greater fast GFR decline in severe-ill patients, such as mechanical ventilation requirement, or low urinary output?

Did the researchers perform a comparative analysis of eGFR at hospital admission between patients with and without COVID-19? Moreover, within the sample of patients with COVID-19. Was there a comparison of eGFR between patients with rapid loss of renal function and patients without rapid loss of renal function?

Discussion

Researchers should engage in a thorough discussion of our findings that 'patients with fast eGFR decline were more likely to have a higher baseline eGFR.' This intriguing contradiction challenges the existing literature, which suggests that CKD (lower GFR) is associated with a greater predisposition to AKI and that the presence of AKI is associated with progression or may even cause CKD.

Researchers should discuss in more detail the possibility of using CART in clinical practice, especially in ICUs, as patients are more severely ill and have a higher frequency of AKI.

And how useful this tool could be for physicians when assessing these patients.

Reviewer #2: The author firstly used Classification and Regression Trees (CART) to identify risk factors of fast GFR decline in post discharge patients during the COVID-19 pandemic. It showed the strength of utilizing machine learning models in the prediction of disease progression. Here are some minor revision advice:

1.The decision to remove variables with more than 5% missing data might lead to loss of potentially important information. It could be beneficial to explore and justify additional methods to handle missing data, such as multiple imputation.

2.Detail whether the matching was performed one-to-one, one-to-many, or another method, as this impacts the analysis.

3.Describe how balance was assessed post-matching, as this is crucial to ensure that the groups are comparable.

4.The rationale for selecting 500 decision trees and three variables for RF models could be more detailed. Also, discuss the sensitivity of results to different hyperparameter choices.

5.Ensure the selection of influential factors in multivariate LR and CART analyses is clearly rationalized, possibly by exploring whether other thresholds for variable inclusion could impact findings.

6.Ensure consistent use of terms and abbreviations throughout the text. For instance, if "eGFR" is used, avoid switching to "GFR" without explanation.

7.Use consistent formatting for statistical data, such as percentages and p-values, to maintain uniformity.

6. PLOS authors have the option to publish the peer review history of their article (what does this mean?). If published, this will include your full peer review and any attached files.

Reviewer #1: **Yes: **Miguel Angelo Goes, MD, PhD, FASN

Reviewer #2: No

---

## [Author Response · Author response to Decision Letter 0]

12 Dec 2024

please see the attached Author Response letter file

---

## [Editor Report · Decision Letter 1]

2 Jan 2025

Classification and Regression Trees analysis identifies patients at high risk for kidney function decline following hospitalization

PONE-D-24-49407R1

Dear Dr. KORAISHY,

We’re pleased to inform you that your manuscript has been judged scientifically suitable for publication and will be formally accepted for publication once it meets all outstanding technical requirements.

Kind regards,

Keiko Hosohata, Ph.D.

Academic Editor

PLOS ONE
---

## [Editor Report · Acceptance letter]

22 Jan 2025

PONE-D-24-49407R1 

PLOS ONE

Dear Dr. Koraishy, 

I'm pleased to inform you that your manuscript has been deemed suitable for publication in PLOS ONE. Congratulations! Your manuscript is now being handed over to our production team.

Kind regards, 

on behalf of

Dr Keiko Hosohata 

Academic Editor

PLOS ONE